# Increased Threshold and Reduced Firing Rate of Auditory Cortex Neurons after Cochlear Implant Insertion

**DOI:** 10.3390/brainsci12020205

**Published:** 2022-01-31

**Authors:** Elie Partouche, Victor Adenis, Dan Gnansia, Pierre Stahl, Jean-Marc Edeline

**Affiliations:** 1Paris-Saclay Institute of Neurosciences (Neuro-PSI), CNRS UMR 9197, Universite Paris-Saclay, 91400 Saclay, France; elie.partouche@universite-paris-saclay.fr (E.P.); victor_adenis@meei.harvard.edu (V.A.); 2Department of Scientific and Clinical Research, Oticon Medical, 06224 Vallauris, France; dagn@oticonmedical.com (D.G.); pisl@oticonmedical.com (P.S.)

**Keywords:** cochlear implant, primary auditory cortex, multi-unit recordings, frequency response area, guinea pig

## Abstract

The cochlear implant (CI) is the most successful neuroprosthesis allowing thousands of patients with profound hearing loss to recover speech understanding. Recently, cochlear implants have been proposed to subjects with residual hearing and, in these cases, shorter CIs were implanted. To be successful, it is crucial to preserve the patient’s remaining hearing abilities after the implantation. Here, we quantified the effects of CI insertion on the responses of auditory cortex neurons in anesthetized guinea pigs. The responses of auditory cortex neurons were determined before and after the insertion of a 300 µm diameter CI (six stimulating electrodes, length 6 mm). Immediately after CI insertion there was a 5 to 15 dB increase in the threshold for cortical neurons from the middle to the high frequencies, accompanied by a decrease in the evoked firing rate. Analyzing the characteristic frequency (CF) values revealed that in large number of cases, the CFs obtained after insertion were lower than before. These effects were not detected in the control animals. These results indicate that there is a small but immediate cortical hearing loss after CI insertion, even with short length CIs. Therefore, efforts should be made to minimize the damages during CI insertion to preserve the cortical responses to acoustic stimuli.

## 1. Introduction

Up to now, very few attempts have been made to quantify the central hearing loss induced by the insertion of cochlear implants. This is of importance because the indication for cochlear implantation has shifted over the last decade: from being only proposed for profound bilateral deafness, it now includes patients with residual hearing at low frequencies but with poor speech understanding with hearing aids. In fact, thirty percent of these patients lose residual hearing after cochlear implantation [1,2,3,4,5]. To preserve residual hearing, a key factor seems to be the control of the mechanical trauma during the implant insertion, which can lead to basilar membrane rupture, electrode translocation from the scala tympani to the scala vestibuli, and spiral lamina fracture. Modelling studies and meta-analyses have tried to identify the critical factors that allow hearing preservation [6,7,8]. Even when local drug delivery has been proposed for otoprotection [9,10], it seems that the stiffness and diameter of the electrode array are crucial: with a thinner and softer electrode array the damage is reduced [1,11]. Recently, two studies have directly, or indirectly, evaluated the hearing loss produced by cochlear implant insertion. Using the auditory brainstem responses (ABRs) as a global measure to assess hearing loss, Drouillard and colleagues quantified the increase in the threshold induced by the insertion of electrode arrays of different diameters and stiffness [12]. It appeared that the larger the array diameter and the stiffer the array, the larger the hearing loss detected from ABRs. More precisely, with the small electrode array (0.3 mm), the ABR threshold increase was 5–12 dB between 8 and 32 kHz from 7 to 30 days post-implantation. With a larger and stiffer electrode array (0.4 mm) the increase in the ABR threshold was 26–44 dB between 8 and 32 kHz from 7 to 30 days post-implantation. In addition, a recent study showed that even CI insertions performed with a motorized device increased the threshold of the compound action potential of the auditory nerve (CAP) in the middle and high frequencies [13]. 

However, a study performed in the Inferior Colliculus (IC) has claimed that the mean threshold of IC neurons was unchanged after the insertion of a cochlear implant (Figure 2B, in Sato and colleagues 2016 [14]). In this study, it was also mentioned (in Materials and Methods, on page 55) that “the ABR threshold measurement confirmed the absence of significant hearing loss due to cochleostomy”, despite the fact that it was reported that drilling the cochlea for the cochleostomy can generate noise levels (130 dB) which can themselves produce acoustic trauma [15,16]. So far, it remains unknown to what extent the ABR threshold correlates with the activity recorded at more central levels of the auditory system or with the behavioral threshold. In two studies, we observed that effects observed on cortical thresholds were larger than those detected based on ABR thresholds. This was the case in a study on aged guinea pigs: there was on average a 30 dB difference between the ABR threshold and the cortical threshold [17]. More recently, we observed that for 21-month-old rats exhibiting less than 15 dB of hearing loss based upon ABR measures, the cortical threshold was increased by 15–25 dB, suggesting higher threshold changes at the cortical than at the brainstem level [18]. 

In the present study, we aimed to quantify the effects of cochlear implantation on the threshold and the evoked firing rate of auditory cortex neurons. This is of importance because the cortical thresholds were shown to be related with the behavioral thresholds in several species (rat: [19] ; Monkey: [20]). We determined the frequency response area (FRA) of neurons in the primary auditory cortex (AI) before and after the insertion of a 300 µm diameter cochlear implant that included six stimulating electrodes (total length of the array 6 mm, distance inter-electrode 1 mm center-to-center). From the FRA, the threshold and the evoked firing rate were determined at the characteristic frequency before and after cochlear implant insertion. The changes observed for these parameters in the implanted animals were compared with those obtained in the control animals where cortical recordings were obtained at least 30 min apart.

## 2. Materials and Methods

### 2.1. Subjects

The subjects used in this study were pigmented guinea pigs (*Cavia Porcellus*) from 6 to 18-months-old and weighing between 700 and 1150 g. The animals had access to food and water ad libitum. They had a heterogeneous genetic background and came from our own breeding colony, which was regularly checked by accredited veterinarians from the Essonne District. All experiments were conducted in accordance with the guidelines established by the European Communities Council Directive (2010/63/EU Council Directive Decree), which are similar to those described in the Guidelines for the Use of Animals in Neuroscience Research of the Society of Neuroscience. The protocol was approved by the ethical committee Paris-Sud and Centre (CEEA N°59, project 2020-20). 

### 2.2. Audiogram

The animals’ audiograms were determined 2–3 days before the experiment by testing auditory brainstem responses (ABR) under isoflurane anesthesia (2.5%). The ABR were differentially recorded via two sub-cutaneous electrodes (SC25, Neuro-Services), one on the vertex and the second behind the mastoid bone, next to the tympanic bulla. The ground electrode was placed in the neck. A dedicated interface and associated software (Otophylab/RT Lab, Echodia, France) allowed us to (i) present sounds monaurally, in close field at specific frequencies with a miniaturized speaker (Knowles Electronics, Itasca, IL, USA) equipped with a 17 mm polyethylene tube, which could be inserted into the animals’ ear canal, and (ii) record the voltage between the two recording electrodes. The signal was filtered (0.2–3.2 kHz, sampling rate 100 kHz) and waveforms were averaged (500 waveforms for each stimulus intensity). The ABR thresholds (decibel SPL, dB) were determined as the lowest level (0 dB SPL = 20 µPa) at which a clear wave III could be observed in the ABR. The animals were tested with pure tones from 0.5 kHz to 32 kHz with octave steps (tone burst, 6 cycles at the plateau, and 2 cycles for the rising and falling slope) presented to one ear at intensities ranging from 80 to −10 dB SPL. All the guinea pigs were adult and some of them displayed modest hearing loss (20–30 dB in the worse cases) corresponding to their age [13,17]. However, we required a threshold of at least 35 dB at 16 kHz for the animals to be included in the experiments. In total, the results presented here came from 43 animals.

### 2.3. Cortical Surgery and Cochlear Implantation

The cochlear implantation was performed under urethane anesthesia (1.2 g/kg, i.p.), supplemented by lower doses (~0.4 g/kg i.p.) when reflex movements were observed after pinching the hind paw. Each animal was initially placed in a stereotaxic frame for the first part of the surgery, the craniotomy. A heating blanket allowed the maintenance of the animal’s body temperature at around 37 °C. After injection of a local anesthetic on the skull (Xylocaine 2%, s.c.), the skin was opened, and the temporal muscles were retracted. The skull was cleaned and dried and three stainless steel screws were threaded into burr holes in the calvarium to anchor a miniature socket embedded in dental acrylic. A craniotomy was performed on the temporal bone, 5 mm behind the Bregma on the rostro-caudal axis and the opening was 8–10 mm wide (as in [18,19,20]). The skin behind the right pinna was opened and the tympanic bulla was exposed. The bulla was opened under binocular control with a 2 mm cutting burr (mounted on a surgical drill) and the cochlea orientation was determined based on anatomical landmarks (round window). A cochleostomy was performed by hand around 1–1.5 mm below the round window with a 0.4 mm diameter trephine. The sterilized electrode-array (300µm in diameter) was a shortened version of the EVO electrode array used by Oticon Medical/Neurelec: it was composed of 6 ring-shaped Platinum–Iridium electrode contacts of the same diameter for a 0.0046 mm^2^ surface, with an inter-electrode distance measured center-to-center of 1 mm (total length of the array 6 mm). The electrode-array and the ground electrode, attached to the miniature socket, were secured to the dental acrylic. The ground electrode was inserted below the skin between the scapulae and the electrode-array was placed in front of the opened tympanic bulla. The electrode-array was inserted into the right scala tympani. A visual confirmation of the number of electrodes inserted within the cochlea was made by direct observation through a binocular microscope. In all cases, only five electrodes were inside the cochlea with the sixth on the edge of the cochleostomy because the array diameter (300 µm) prevented us from inserting the array beyond the first turn and half in the guinea pig cochlea, which roughly corresponded to frequencies between 8 and 12 kHz [21,22]. In all cases, the electrodes’ impedances were checked (around 2000–3500 Ω) to confirm the number of electrodes properly inserted in the cochlea. At the end of the data collection, animals were euthanized by a lethal injection of Dolethal (200 mg/kg).

### 2.4. Recording of Auditory Cortex Neurons

During the recording session, performed also under urethane anesthesia, multiunit activity (MUA) was recorded in the primary auditory cortex (A1). The methods and data acquisition were exactly the same as in our previous studies [18,23,24,25]. A 16-electrode array (ø: 33 µm, <1 MΩ), composed of two rows of 8 electrodes separated by 1000 µm (350 µm between electrodes of the same row), was inserted in A1 perpendicular to the cortical surface to record multi-unit activity in layer III/IV (depth: 500/600 µm). A small silver wire (ø: 200 µm), used as the ground, was inserted between the temporal bone and the dura matter on the ipsilateral side. The location of the primary auditory cortex was estimated based on the pattern of vasculature observed in previous studies [19,20,21,26,27,28,29,30]. The raw signal was amplified 10,000 times (TDT Medusa). It was then processed by an RX5 multichannel data acquisition system (TDT). The signal recorded from each electrode was filtered (610–10000 Hz) to extract the MUA. The trigger level was set for each electrode to select the largest action potentials from the signal. On-line and off-line examination of the waveforms indicated that the MUA recorded was composed of a few shapes of action potentials (generally less than 6) from neurons in the vicinity of the electrode. For each experiment, the position of the electrode array was set in such a way that the two rows of eight electrodes sampled neurons responding from low to high frequency when progressing in the rostro-caudal direction (see examples of tonotopic gradients recorded with such arrays in Figure 1 [22] and in Figure 6A [25]). 

### 2.5. Experimental Protocol

Fifteen minutes after the stabilization of the 16-electrode array in the auditory cortex, pure tones covering eight octaves (0.14–36 kHz) were presented at 75 dB SPL in a random order at a rate of 4.15 Hz. On-line visualization of the spectro-temporal receptive fields (STRFs) allowed for an evaluation of the stability of the multi-unit responses below each electrode. Once the stability was satisfactory for the set of responsive electrodes, the frequency response area (FRA) was determined by presenting pure tones (0.14–36 kHz) from 75 to 5 dB SPL (5 dB steps, random order) at a rate of 2 Hz. Each tone was presented eight times at each intensity and each frequency. Pictures of the electrode array placement in the auditory cortex were taken under several angles to visualize the exact location of the 16-electreode array relative to the cortical vasculature. The electrode array was then removed to insert the cochlear implant in the animal cochlea (see above). Once the cochlear implant was secured to the bulla, the electrode array was placed back at the same cortical location with the help of the pre-implantation pictures, which allowed the neuronal responses in the same cortical columns to be investigated. Based upon an examination of the pictures obtained before cochlear implant insertion by the two co-authors, the precision of the location of the 16-electrode array on the cortical surface could be estimated to be less than 100µm. The electrode array was lowered at the same depth than before the cochlear implant insertion. After waiting 15 min for the stabilization of the array in the cortex, pure tones were presented at 75 dB, and the complete FRA was re-determined between 5 and 75 dB. For the control animals (*n* = 3), the 16-electrode array was removed, then placed back 30 min later to obtain an estimation of the variability of the cortical responses obtained by re-sampling the same cortical sites twice 30 min apart. In the case of these control animals, the bulla was not opened and the cochleostomy was not performed. The 16-channel cortical matrix was placed back at the same cortical position three to six times (with at least 30 min between each re-positioning) to collect as much data as possible on these control animals (according to the 3R rules for animal experimentation). In each case, comparisons were made between the FRA parameters derived from the two successive cortical matrix positions.

### 2.6. Quantification of the Cortical Responses

The analyses of the cortical responses included quantifying the evoked responses (spikes/sec) at 33 frequencies going from 0.14 kHz to 36 kHz for intensities ranging from 5 dB to 75 dB SPL (5 dB steps, random order). The evoked firing rate was quantified over 50 ms after tone onset. For each recorded electrode, significant evoked responses above spontaneous activity (+6 sem) were determined at each frequency and intensity leading to a frequency response area (FRA) for each recording. The frequency providing significant responses at the lowest intensity was defined as the characteristic frequency (CF). At the CF, the lowest sound intensity providing a significant evoked response was defined as the threshold.

### 2.7. Statistical Analysis

In each group of animals (implanted and control), after checking the normality of the distributions (with a Shapiro–Wilk test) and the variance homogeneity (with an F-test), the changes in CFs values, in the threshold and in the evoked firing rate between pre- and post-implantation were tested by paired t-tests. A Benjamini–Hochberg correction was applied to account for multiple testing. Unpaired *t*-tests (Mann–Whitney and Welch’s correction) were used for between-group comparisons. Tests were performed with GraphPad Prism (version 9.3.1).

## 3. Results

Data were obtained from 43 guinea pigs: 40 were implanted animals and 3 were used as the controls. From these animals, we obtained 576 cortical multi-unit recordings from the primary auditory cortex in the implanted animals and 167 recordings in the control animals (see Methods for details). The effects of inserting a cochlear implant on the responses of auditory cortex neurons were assessed by comparing for each recording the parameters obtained for the pre- and post-insertion FRAs. These effects were contrasted with those obtained in the control animals.

### 3.1. Consequences of Cochlear Implantation on Frequency Response Areas

Figure 1A–C represent three sets of frequency response areas (FRAs) simultaneously obtained at eight positions of the tonotopic map using a linear array of electrodes in the primary auditory cortex (A1). In the three cases, at the first insertion of the electrode array (blue curves), a smooth progression of the neurons’ characteristic frequency (CF) was clearly detected from low frequencies in the rostral positions (CH1) to high frequencies in the more caudal positions (CH8).

Figure 1A,B display examples of the FRAs obtained in two different implanted animals before (blue curves) and after (red curves) the insertion of the cochlear implant. Figure 1A shows a typical example where an increase in the threshold at the CF was observed for seven/eight cortical sites. Figure 1B shows an atypical case where the effect was a shift of the entire FRA and of some CFs toward low frequencies, with or without an increase in the threshold.

For the control animals, the impact of the electrode’s insertion on the CF sampling was determined by removing the cortical matrix after its first penetration in the cortex (blue curves in Figure 1C) and re-inserting it 30 min later at the same location as assessed visually under the microscope by two co-authors (E.P. and V.A.). After reaching the same depth as before (600 µm below pia), the FRAs of the auditory cortex neurons were re-determined (Figure 1C, red curves for each electrode). For each electrode there was little or no variation in the CF value and in the threshold at the CF.

These control data suggest that re-inserting the electrode array in the cortex at the same location (based upon the patterns of vasculature and based on a 100 µm precision) allowed the same cortical columns to be sampled compared with the first FRA determination.

### 3.2. Quantification of the CF Values on the Whole Database

Based on the FRAs, the CF value of each cortical site was automatically quantified by a MatLab script (see Methods for details) before and 30 min after cochlear implant insertion. 

The scattergram displayed in Figure 2A presents the CF values before (*x*-axis) versus these values after insertion (*y*-axis). In a large number of cases, the CF values obtained after insertion were lower than before the insertion (dots below the diagonal line). Statistical analysis confirmed that, on average (over the 33 frequencies), the CF values were lower after than before insertion (paired *t*-test, *p* < 0.0001). In the low frequencies (CF < 4.5 kHz) there was, on average, little change in the CF value (paired t-test, *p* = 0.3605), whereas in the middle (4.5 ≤ CF < 12.7 kHz) and high frequencies (CF ≥ 12.7 kHz), there were relatively large decreases in the CF value (paired t-tests, *p* = 0.0099 and *p* = 0.0054 in the middle and high frequencies, respectively). The quantification of these CF shifts is presented on the bar graphs below the scattergrams (Figure 2C). On average, the change in CF values (mean ± SEM) was 0.12 ± 0.06 octaves in the low frequencies, −0.28 ± 0.05 octaves in the middle frequencies, and −0.82 ± 0.07 octaves in the high frequencies (negative values indicate lower CF values after CI insertion).

In contrast, when the same analysis was performed on the FRAs obtained in the control animals (Figure 2B), there were only a few cases of lower CFs at the second insertion of the cortical matrix. Statistical analysis confirmed that in the control animals the CF values did not differ at the first and second insertion (paired *t*-test, *p* = 0.3879). In fact, as shown by the scattergram (Figure 2B), in most of the cases, there was a good match between the CF values obtained at the first and second insertion of the cortical electrode matrix (dots on, or around, the diagonal line). The CF values did not differ between two successive FRA determinations in the high and middle frequencies (*p* = 0.5855 and *p* = 0.4118, respectively) but they were slightly lowered in the low frequencies (*p* = 0.0136). The quantification of the CF shifts in the control animals are presented on the bar graphs below the scattergrams (Figure 2D). This shows that in the middle (4.5 ≤ CF < 12.7 kHz) and high frequencies (CF ≥ 12.7) there was on average no change in the CF value, and only small changes in the low frequency (CF < 4.5 kHz). On average, the change in CF values was −0.28 ± 0.08 octaves in the low frequencies, 0.007 ± 0.09 in the middle frequencies, and 0.004 ± 0.03 octave in the high frequencies. 

In addition, between-group comparisons indicated significant changes between the implanted and control groups (unpaired t-tests with Welch correction; *p* = 0.0028, *p* < 0.0101 and *p* < 0.0001 in the low, middle, and high frequencies, respectively). 

In summary, cochlear implant insertion led to an immediate shift of the CF values toward lower values in the high and middle frequencies. This effect was not observed in the control animals.

### 3.3. Group Data for the Changes in the Acoustic Threshold 

Based on the FRAs, we systematically quantified the changes in the acoustic threshold obtained at each of the tested frequencies in the implanted and control animals. 

The curve displayed in Figure 3A shows, for the implanted animals, the mean difference (±sem) between the post- and the pre-threshold obtained at each frequency from 0.96 to 36 kHz. On average, for the low frequencies (<4.5 kHz), the thresholds were slightly, but not significantly different before than after cochlear implant insertion (paired *t*-tests with Benjamini–Hochberg correction, all adjusted *p*-values > 0.05, except for *p*-value (3.8 kHz) = 0.01). Then starting from 6.35 kHz up to 36 kHz, the thresholds were significantly higher after insertion than before insertion (paired *t*-tests with Benjamini–Hochberg correction, all adjusted *p*-values < 0.001), leading to positive threshold shifts of about 10–15 dB in the high frequency range (from 15 to 36 kHz). The inset in Figure 3A shows the mean threshold change in the low, middle, and high frequencies: on average the increase in the threshold in the low frequency (CF < 4.5 kHz) was 0.66 ± 1.56 dB, it was 4.11 ± 1.21 dB in the middle (4.5 ≤ CF ≤ 12.7 kHz), and it was 11.3 ± 1.90 dB in the high frequency (CF ≥ 12.7 kHz). 

In contrast, in the control animals, the quantification the threshold changes between two successive insertions of the cortical matrix showed that there was no significant change in threshold in the low, middle, and high frequencies (paired *t*-tests with Benjamini–Hochberg correction, all *p*-values > 0.05). The inset in Figure 3B shows the mean threshold change in the low, middle, and high frequencies: on average the threshold change in the low frequency (CF < 4.5 kHz) was −0.16 ± 1.88 dB, it was 2.91 ± 1.61 dB in the middle (4.5 ≤ CF ≤ 12.7 kHz), and it was 2.96 ± 2.11 dB in the high frequency (CF ≥ 12.7 kHz). Between-group comparisons for the values of threshold shifts indicated that the threshold changes differed in the high frequencies (unpaired *t*-test, *p* < 0.001) but did not differ in the low and middle frequencies (unpaired t-tests, *p* = 0.16 and *p* = 0.24, respectively). 

In summary, cochlear implant insertion led to an immediate increase in acoustic threshold in the high frequencies. This effect was not observed in the control animals.

### 3.4. Group Data for the Evoked Firing Rate

Next, we systematically quantified the changes in the evoked firing rate. This was performed at the suprathreshold level (75 dB) in an attempt to quantify these changes in the evoked firing rate independently of the increase in threshold. This was quantified for each of the tested frequencies in the implanted and control animals. 

For the implanted animals, the curve displayed in Figure 4A shows the mean (± sem) difference between the post- and the pre-evoked firing rate obtained at each frequency from 0.96 to 36 kHz. For most of the high frequencies (from 18 to 36 kHz), the difference in the evoked firing rate was largely negative, indicating that the evoked firing rate was much lower after the insertion of the implant than before. This decrease in the evoked firing rate was significant from 21 kHz to 36 kHz (paired *t*-test with Benjamini–Hochberg correction, all adjusted *p*-values < 0.001). Surprisingly, an increase in the evoked firing rate was detected in the low and mid frequencies, an effect that was significant from 2.7 kHz to 5.35 kHz (paired *t*-tests with Benjamini–Hochberg correction, all adjusted *p*-values < 0.001) and reached its peak at 3.8 kHz. The inset in Figure 4A confirms that on average, there was an increase in the evoked firing rate in the low (23.35 ± 10.98 spikes/sec) and middle frequencies (10.88 ± 6.70 spikes/sec) and a large decrease in the evoked firing rate in the high frequencies (−31.69 ± 8.72 spikes/sec). 

In contrast, in the control animals, there was no change in the evoked firing rate in the high frequency range for all tested frequencies (paired *t*-tests with Benjamini–Hochberg correction, all *p*-values > 0.05 from 12.7 to 36 kHz). On average, the decreases in the evoked firing rate observed in the low and mid frequency ranges were never significant (paired t-test with Benjamini–Hochberg correction, all adjusted *p*-values > 0.05). The inset in Figure 4B confirms that on average, there was a slight decrease in the evoked firing rate in the low (−25.39 ± 13.43) and middle frequencies (−18.85 ± 12.19) and no change in the evoked firing rate in the high frequencies (−2.65 ± 14.25). The between-group comparisons were performed using Mann–Whitney tests since the distributions of the evoked firing rate changes significantly differed from normal ones. They indicated that the evoked firing rate changes differed between the two groups in the three frequency ranges (low frequencies: *p* < 0.0001; middle frequencies: *p* = 0.0022; and high frequencies: *p* = 0.0379). 

Based on the increase in the acoustic threshold and on the decrease in the evoked firing rate at 75 dB described in the last two sections, one can suspect that the intensity coding is altered after CI insertion. More precisely, if the acoustic threshold is higher and the evoked firing rate is reduced at 75 dB, this can potentially reduce the dynamic range of the cortical evoked responses and also produce stepper rate-level functions. However, an examination of the clearest cases of increase in the threshold and decrease in the evoked firing rate at 75 dB from our database failed to reveal changes in the slope of the rate-level functions at the CF. Three examples presented in Figure 5 indicate that the rate-level functions obtained before and after CI insertions were parallel without obvious slope changes. 

In summary, inserting a cochlear implant generates an immediate decrease in the evoked firing rate in the high frequency range and a rebound in the evoked firing rate in the middle and low frequencies. In the control animals, there was no change in the evoked firing rate in the low, middle, and high frequencies.

## 4. Discussion

The results presented here point out that several alterations in the responses of primary auditory cortex neurons can be detected immediately after inserting a cochlear implant in the guinea pig cochlea. In a large proportion of cases, there was a shift in the CF frequency toward lower frequencies for the neurons with their initial CF in the high frequency range. This effect was accompanied by an increase in the threshold and a reduced evoked firing rate at suprathreshold intensities (75 dB) in the high frequencies. Figure 6 summarizes most of the effects observed in the implanted animals. Before discussing the clinical implications of these findings, some potential mechanisms will be presented.

### 4.1. Potential Neuronal Mechanisms Underlying the Consequences of CI Insertion

Inserting a CI in the cochlea, even when performed by expert surgeons, necessarily produces cochlear damage [33]: the loss of perilymph, the inflammatory reactions triggered by the opening of the bulla and of the scala tympani, and the insertion of the CI are physiological events which all impact on the processing of auditory information by the cochlea [34,35,36]. In addition, the presence of the CI implant itself within the scala tympani might prevent the basilar membrane from vibrating and resonating normally at sound presentations, or even increase the stiffness of the round window [7]. Because of these physiological alterations, an atraumatic insertion appears as a difficult challenge and limited damage seems difficult to avoid. 

In our study, we used a relatively short CI device (6 mm in total length), which could be inserted until the end of the first and half turn of the guinea pig cochlea. From the tonotopic organization of the guinea pig cochlea it seems that this corresponds to the 8–12 kHz frequency range [22]. As a consequence, in our conditions, the damage generated by the CI insertion should have been in the high and mid-frequency domains based upon the estimation of the length of guinea pig cochlea. The effects described here share some similarities with what has been reported after mechanical lesions of the cochlea or after acoustic trauma. The immediate consequences of partial cochlear lesions have been assessed at several levels in the auditory system [37,38,39,40,41,42,43,44]. At the cortical level, the post-lesion thresholds were largely elevated but there were “residual responses” in the absence of cortical responses at the original CF [37,38]. Compared to the pre-lesion responses, these residual responses displayed CF shifted toward low frequencies, potentially because high frequency auditory nerve fibers have low frequency “tails”. 

In addition, it is well known from intracellular and whole-cell recordings that auditory cortex neurons integrate very broad ranges of subthreshold inputs [31,45,46,47], much larger than the classical receptive fields that are described from extracellular recordings [48,49,50,51,52]. As a consequence, at a given cortical site, when the dominant thalamic inputs are lost (or reduced) after peripheral damage, the cortical neurons still receive non-dominant (non-CF) inputs, usually from lower frequencies and they exhibited both increased thresholds and new CFs shifted toward lower frequencies (see diagram in Figure 6C–E). The cortical receptive fields observed here after CI insertion most likely reflect, at least partially, residual responses when damages are made in the high frequency range of the cochlea due to CI insertion in the first and half turn of the cochlea. 

Obviously, one of the pitfalls of the present experiment was that we assessed the effects of CI insertion after removing and re-positioning the cortical electrode array. This is why it was crucial to collect control recordings in the same conditions without inserting a CI in the animal’s cochlea. These recordings displayed no increase in the thresholds, no decrease in the evoked firing rate and, on average, no shift of their CF after re-positioning the cortical electrodes, thus indicating that most of the effects detected here in the implanted animals resulted from damage induced by CI insertion.

### 4.2. Clinical Implications

Initially, the cochlear implant (CI) was a prosthetic device designed for severe-to-profound sensorineural hearing loss, which explains the lack of interest for detecting potential hearing loss generated by CI insertion. However, over the last few years, many implanted subjects had residual hearing that needed to be preserved during CI insertion. Therefore, hearing preservation, especially in low frequencies, became an important topic since it is suspected that it will help music and speech perception and benefit the overall quality of life [6]. 

To reinforce the use of this low frequency residual hearing, electro-acoustic stimulation (EAS), combining both electrical stimulation and bone-conduction advantages, has been developed to increase speech understanding. Our results indicate that the use of short CI devices allows the preservation of low frequencies representation at the cortical level, both in terms of the threshold and in terms of the evoked firing rate. However, it is important to keep in mind that the present data were obtained immediately after cochlear implant insertion, a situation which considerably differs from the tests performed months after surgery and during the patient’s life. In fact, hearing preservation during surgery remains a major step because the residual hearing can either stay stable or decrease over months [2,6,8], suggesting that the hearing loss detected immediately post-surgery is most likely the starting point from which effects emerge in the following months. 

In fact, between 30 and 55% of Electro Acoustic Stimulation CI patients lose 30 dB or more of their residual low-frequency hearing within months of implantation [3,4,5,33], and investigations should be made to characterize their long-term hearing loss [2]. Efforts are now being made to design electrode arrays that will reduce the hearing loss on these subjects. For example, a new thin lateral wall electrode array (HiFocus SlimJ) was designed based on μCT studies of human cochlea anatomy [53]. In the literature, modelling studies suggest that post-implantation hearing loss can occur in low frequencies because of an increase in round window stiffness [7,54]. This potentially occurs when the CI is inserted via the round window but also if fibrotic tissue develops on the round window after surgery. However, in the case of a cochleostomy (as in our case), the mechanical damage might be limited to the frequency range of the most apical location reached by the CI in the scala tympani.

Several recent studies have also pointed out that compared with a manual insertion, intracochlear trauma could be reduced with array insertion performed with an optimal axis and controlled by a robot-based device applying a constant motorized insertion speed [35,36]. Many factors can in fact contribute to reducing cochlear damage such as the electrode stiffness, the maximal peak of insertion force measured during insertion [12], or the application of pharmacological agents [9]. 

Compared to the classical ABR measurements where only thresholds can be determined, the main advantage of cortical recordings is that they probably provide a neural index closer from perception [19,20] and that they can indicate more precisely which frequency was impacted. Thus, the animal model we have proposed here can be useful for testing the potential consequences of different types of CI, or different insertion strategies in human patients.

## 5. Conclusions

Based on the existing human literature, it seems that, even when performed by expert surgeons, CI insertion by cochleostomy produces cochlear damage which leads to increases in pure tone audiometry in frequency ranges corresponding to the CI location in the scala tympani, a finding confirmed by our cortical recordings. In fact, the increase in pure tone audiometry observed in humans after CI insertion [2] can potentially result from the increase in the cortical threshold described in our animal model. Therefore, all possible efforts should be made (i) to minimize cochlear damage during implantation (for example with the use of robotized devices [55,56]) and (ii) to attenuate the cortical increase threshold occurring over the long term after insertion.

## Figures and Tables

**Figure 1 brainsci-12-00205-f001:**
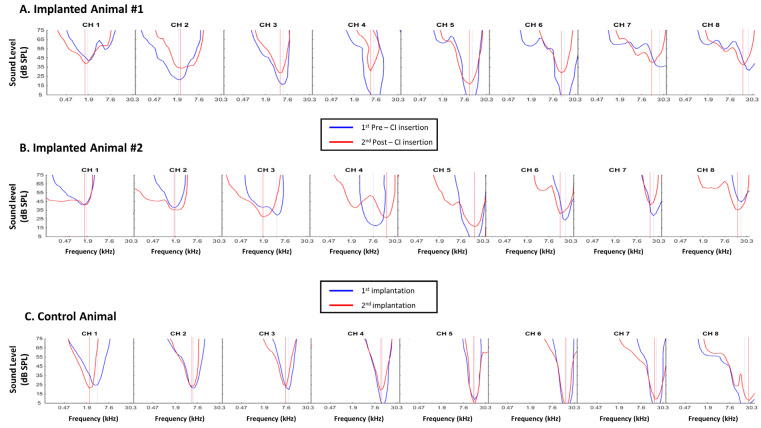
Individual Frequency Response Areas (FRA) obtained before and after cochlear implant insertion (**A**,**B**) or after two consecutive implantations of the cortical electrodes matrix in a control animal (**C**). For each row in (**A**–**C**), the curves represent the FRA obtained from 8 different electrodes aligned in the rostro-caudal axis with CH1 located in the most rostral part and CH8 located on the most caudal part of the primary auditory cortex. Each curve was automatically generated from the evoked responses to 33 frequencies (from 0.14 kHz to 36 kHz) presented at 5 to 75 dB and delineates the range of frequencies and intensities that elicited significant evoked responses. (**A**,**B**) Each blue curve corresponds to the FRA obtained for the 8 electrodes before the cochlear implant insertion and each red curve corresponds to the FRA obtained for the same 8 electrodes after the cochlear implant insertion. In all cases, the CF values obtained before and after insertion are indicated by the dashed vertical lines (blue: before; red: after). In many cases (9/16 in these two examples), the CF values shifted toward lower frequencies and there was also an increase in the threshold at the CF frequency (especially in (**A**)). (**C**) Each blue curve corresponds to the initial FRA obtained for a set of 8 electrodes and each red curve corresponds to the FRA obtained after removing the cortical matrix and re-positioning it at the exact same location 30 min later. In all cases, the CF values obtained at the 1st and 2nd placement of the cortical matrix are indicated by the dashed vertical lines (blue: 1st; red: 2nd). In most of the cases (5/8 in these example), the CF values were the same at the 1st and 2nd placement of the matrix and there was also little increase in the threshold at the pre-insertion CF frequency.

**Figure 2 brainsci-12-00205-f002:**
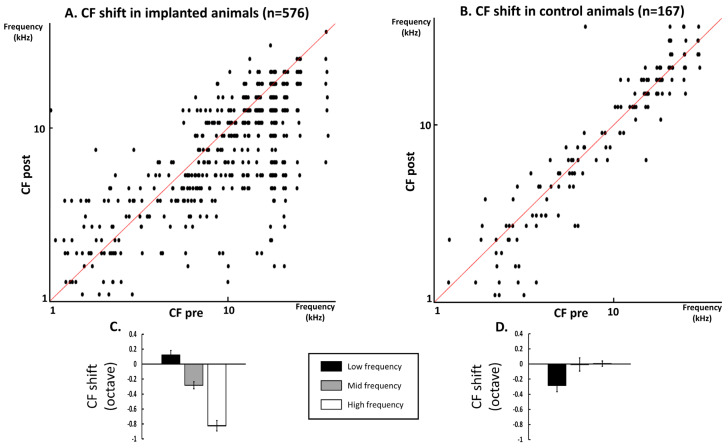
Quantification of the shift in CF values in the cochlear implanted animals (**A**,**C**) and in the control animals (**B**,**D**). A. Scattergrams representing the CF values obtained before cochlear implant insertion (abscissa) versus the CF values obtained after cochlear implant insertion (ordinates). Note that in a large majority of the cases, the CF values were lower after cochlear implant insertion (dots below the diagonal line). (**B**) Scattergrams representing the CF values obtained at the first placement of the cortical matrix of electrodes (abscissa) versus the CF values obtained at the second placement of the cortical matrix of electrodes (ordinates). Note that in a large majority of the cases, the CF values were quite similar (dots on or around the diagonal line). (**C**) Quantification of the mean CF shift (±sem) in the implanted animals in the low (0.95–3.8 kHz), middle (4.5–10.7 kHz) and high (12.7–36 kHz) frequency ranges. On average, the changes were 0.15 ± 0.10 octaves in the low frequencies, −0.40 ± 0.06 octaves in the middle frequencies, and −0.79 ± 0.06 octaves in the high frequencies (negative values indicate lower CF after CI insertion). (**D**) Quantification of the mean CF shift (±sem) in the control animals in the low (0.95–3.8 kHz), middle (4.5–10.7 kHz), and high (12.7–36 kHz) frequency ranges. On average, the change in CF values was −0.28 ± 0.08 octaves in the low frequencies, 0.004 ± 0.07 in the middle frequencies, and −0.002 ± 0.03 octave in the high frequencies.

**Figure 3 brainsci-12-00205-f003:**
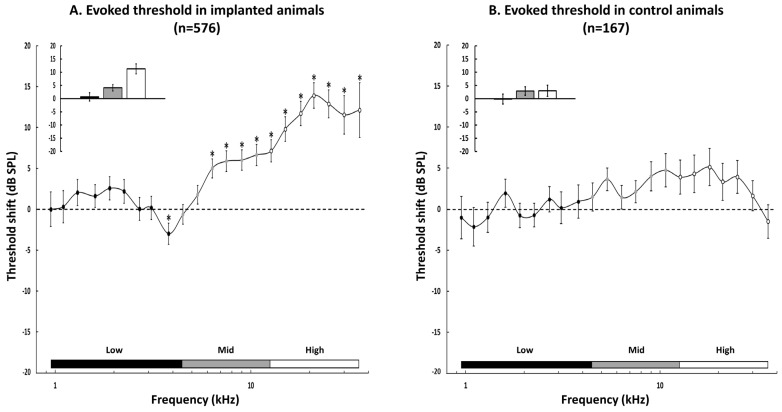
Quantification of the threshold changes in the cochlear-implanted animals (**A**) and in the control animals (**B**). (**A**) At each frequency used to test the Frequency Response Area (FRA), we automatically detected the threshold value before and after cochlear implant insertion and computed the threshold shift as the difference between the post and the pre value. Negative values indicate decreased thresholds and positive values indicate increased thresholds. The threshold changes (mean ± sem) are presented for the low frequencies (0.95–3.8 kHz, black dots), middle (4.5–10.7 kHz, grey dots), and high (12.7–36 kHz, white dots) frequencies. There were non-significant changes in the low frequencies. Starting in the middle frequencies (at 6.35 kHz), the thresholds were increased post-implantation and the maximum increase was around 14 dB. Asterisks represent significant changes post-insertion compared to pre-insertion (assessed by paired t-tests with a *p*-value < 0.05 after a Benjamini–Hochberg correction for multiple testing). The inset shows the mean threshold changes in the low, middle, and high frequencies. (**B**) Same representation as in A with the recordings obtained from the control animals. The threshold values were derived from a FRA collected at the 1st implantation of the cortical electrode matrix, then collected after a second implantation 30 min later. There was no change in the low frequencies and there were some slight increases in middle and high frequencies, but they were not significant. The inset shows the mean threshold changes in the low, middle, and high frequencies: these changes were significantly different from the CI-implanted animals in the high frequencies (see text for details). In A and B, * indicate significant differences between the Pre and Post values of threshold.

**Figure 4 brainsci-12-00205-f004:**
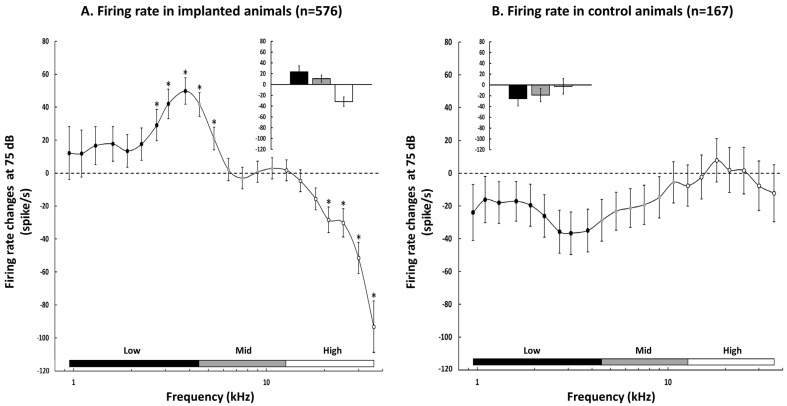
Quantification of the evoked firing rate changes (at 75 dB) in the cochlear-implanted animals (**A**) and in the control animals (**B**). (**A**) At each frequency used to test the Frequency Response Area (FRA), we automatically quantified the evoked responses (at 75 dB) before and after cochlear implant insertion and computed the evoked firing rate change as the difference between the post- and the pre-evoked responses. Negative values indicate decreases in the evoked firing rate and positive values indicate increases in the evoked firing rate. The changes in the evoked firing rate (mean ± sem) are presented for the low frequencies (0.95–3.8 kHz, black dots) and the middle (4.5–10.7 kHz, grey dots) and high (12.7–36 kHz, white dots) frequencies. There were some significant increases in the evoked firing rate at the limit between the low and the middle frequencies (from 2.7 to 5.35 kHz). The evoked firing rates were systematically decreased in the high frequencies; this decrease was significant from 21 to 36 kHz. Asterisks represent significant changes post-insertion compared to pre-insertion (assessed by paired *t*-tests with *p*-value < 0.05). The inset shows the mean threshold changes in the low, middle, and high frequencies. (**B**) Same representation as in (**A**). for the recordings obtained from the control animals. The evoked firing rates were derived from a FRA collected at the 1st implantation of the cortical matrix then after the second implantation 30 min later. There was no significant change in the low and middle frequencies (from 2.5 to 3.1 kHz), and also no significant changes in the high frequencies. The inset shows the mean threshold changes in the low, middle, and high frequencies: these changes were significantly different from the CI-implanted animals in the three frequency ranges (see text for details). In A and B, * indicate significant differences between the Pre and Post values of firing rate.

**Figure 5 brainsci-12-00205-f005:**
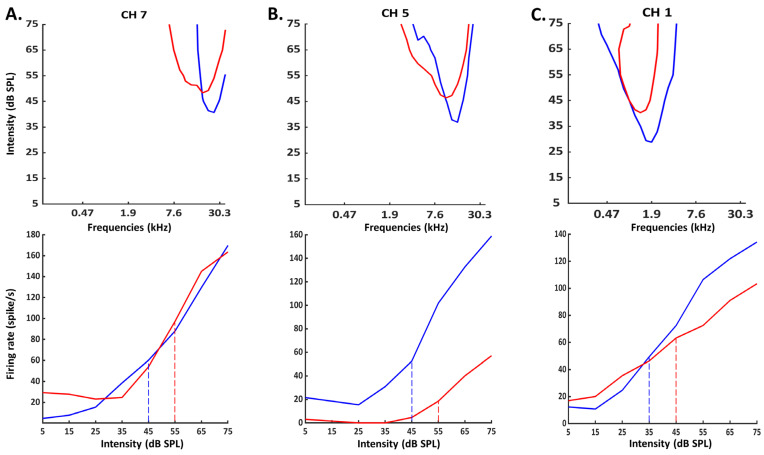
Examples of Frequency Response Areas (FRA) and rate-level functions obtained before and after CI insertion. Top: For each example, the bleu curve represents the FRA obtained before CI insertion and the red curve the FRA obtained after CI insertion. Bottom: For each example, the bleu curve represents the rate-level function obtained before CI insertion and the red curve the rate-level function obtained after CI insertion. (**A**) In this case, the threshold (defined as a response above spontaneous activity plus 6 sem) increased from 45 to 55 dB and there was only a small decrease in evoked firing rate at 75 dB. (**B**) In this case, the threshold (defined as a response above spontaneous activity plus 6 sem) increased from 45 to 55 dB and there was a large decrease in evoked firing rate at 75 dB. (**C**) In this case, the threshold (defined as a response above spontaneous activity plus 6 sem) increased from 35 to 45 dB and there was a decrease in evoked firing rate at 75 dB. Note that in the three cases, there was not increase in the slope of the rate level function after CI insertion: the slope was either similar or slightly smoother after than before CI insertion.

**Figure 6 brainsci-12-00205-f006:**
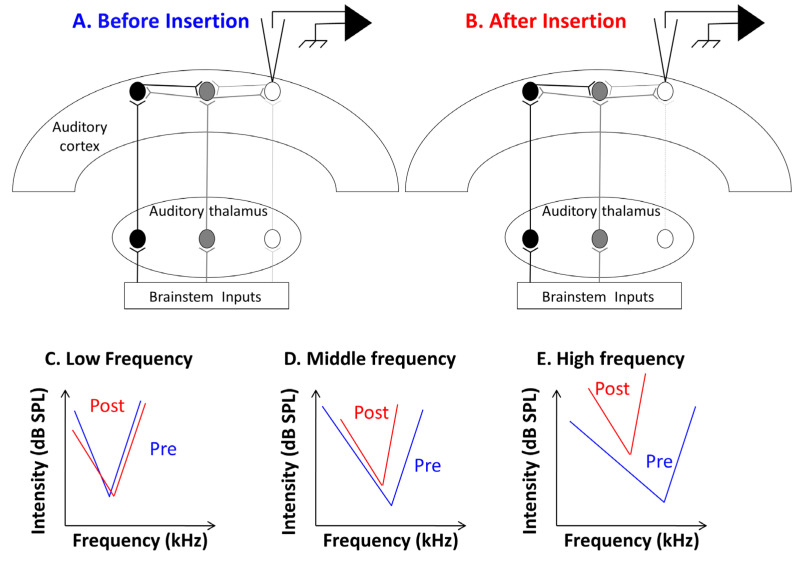
Schematic representation of the potential mechanisms involved in the effects detected after CI insertion. (**A**) In the auditory cortex, a given cortical neuron recorded in the high frequency region, receives thalamic inputs but also cortico-cortical inputs from other cortical regions. For each cortical neuron, the inputs from the auditory thalamus provide the lowest threshold at the CF, but, at suprathreshold level, the cortical inputs also contribute to the frequency response area (FRA, see [31]; review in [32]). (**B**) After cochlear implant insertion, different mechanical damages made by opening the bulla and performing the cochleostomy around the round window generate a partial loss of synaptic inputs in the high frequency range (indicated by the lack of brainstem inputs in the region). This contributes to an increase in the threshold at the pre-insertion CF. The additional cortico-cortical inputs now provide a new CF (in the middle frequency range) with a higher threshold. (**C**–**E**) Schematic diagrams illustrating the consequences on the FRA of cortical neurons in the low, middle, and high frequency range. The low frequency neurons do not show an increased threshold and CF shift (**C**). The middle frequency neurons show some increase in the threshold and some CF shift toward lower frequencies (**D**). The high frequency neurons show a large increase in the threshold and large CF shifts toward lower frequencies (**E**).

## Data Availability

Data presented in this study are available upon request from the corresponding author.

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
