# Peer review of "Increased Threshold and Reduced Firing Rate of Auditory Cortex Neurons after Cochlear Implant Insertion"

_brainsci, 2022, doi:10.3390/brainsci12020205_

Round 1

Reviewer 1 Report

This study evaluates a very useful and important topic: auditory cortical neurons activity after cochlear implantation. It is an animal study very important for the consequences of cochlear implant in human subjects.

I would like to suggest some minor revisions.

The study is methodologically well-conceived and executed.

All part of the article (Abstract, Introduction, Material and Methods, Results, Discussion and Conclusion) are well-conceived and well-structured.

I suggest you to remove the sentences in the page 2 (lines 74-78) because is a discussion part and not an introduction part.

Author Response

Answer: As requested by the reviewer 1, we have removed the last sentence of the introduction.

Reviewer 2 Report

Summary:

In the manuscript entitled: „Increased Threshold and Reduced Firing Rate of Auditory Cortex Neurons after Cochlear Implant Insertion” Partouche and colleagues investigate the effects of surgical measures during cochlear implantation on auditory thresholds as evaluated by multi-electrode recordings from the auditory cortex in anesthetized guinea pigs. The cochlear implants (CI) were inserted close to the round window, i.e. in a tonotopic position associated with higher frequencies in the cochlea. With multi-unit recordings in the auditory cortex, the authors find increased thresholds after CI implantation also in a higher frequency region. Here, also auditory thresholds were increased while stimulus-evoked responses were reduced.

Evaluating hearing loss after CI implantation is of significant clinical relevance as increasingly more patients are implanted with CI that are not completely deaf but – usually – contain residual low-frequency hearing. Further work has shown that the combined electric and acoustic stimulation available to these patients results in a substantially better hearing restoration than providing electrical stimulation alone. The preservation of this low-frequency hearing is thus of outstanding importance.

The article is well written, clearly structured and easy to follow.

I have several concerns that mostly concern how the work is motivated and put into context.

Major points:

  1. Identifying factors that lead to or cause hearing loss are important lines of investigation in CI research. In humans, there is data to suggest that the strongest hearing loss actually occurs in the lower frequency regions. One factor discussed here is potential stiffening of the round window which has been modelled to result in ‘hearing loss’ of as much as 30 dB solely due to biomechanical changes in the cochlea (Elliot et al. 2016, https://doi.org/10.1016/j.heares.2016.08.006). Other work has explored how to directly assess neural health in different frequency regions of the cochlea after CI implantation (Adel et al. 2020; https://doi.org/10.1097/aud.0000000000000910). This latter work also demonstrated threshold increases located at higher frequencies. These, however, were much larger (ca. 30 dB) than the changes presented in the current manuscript.

In my reading, together this might suggest a discrepancy between animal research seeking to identify contributions to hearing loss and predictions for long term preservation where changes seem linked to the actual position of the CI and human results that seem to demonstrate losses most strongly in frequency regions where the CI is not necessarily located.

As the current work is motivated to provide support for clinical work, this should be discussed and be put into perspective. Further also taking into account that the work is based on acute experiments.

  1. The authors argue that their results are in contrast with Sato et al. 2016. However, there is not enough information for the reader to evaluate this. The methods section does not provide sufficient detail on the exact order of surgical events. Sato et al. only compare thresholds before and after CI insertion which – in their case – happens after the IC electrode is in place and after the bullostomy and cochleostomy have been performed. Maybe there really is no ‘significant’ additional change in thresholds after CI insertion in their case? Also, if one checks figure 2B in Sato et al. – as the authors themselves suggest - there is about an 8 to 11 dB shift after implantation for frequencies higher or equal than 13 kHz. The authors also refer to own data that threshold changes can actually increase from the periphery to central stations. Based on the data in the current manuscript and the fact that the CI array in Sato et al. is shorter, I think this actually is a very nice correspondence.

The maximum threshold shifts were observed in the high frequency range and found to be 11 dB on average. This is much smaller than expected? And is quite close to the definition of complete preservation used in some meta-analyses of hearing preservation in humans (< 10 dB).

I thus strongly suggest to tone down those claims and revise the relevant sections in the introduction and discussion.

  1. In general, the methods section could be enhanced to provide the reader with a better understanding of the data.
    • It does not become clear how such a large discrepancy between number of animals in the CI and control group lead to relatively small differences in multi-unit recordings (40 to 3 vs. 576 to 167). The data analysis description suggests that for the controls only the first and second insertion were compared.
    • It does not become totally clear what happened exactly to the control animals: was the bulla opened? Was a cochleostomy performed? This should be spelled out. From the methods I would have thought that no bulla surgery via a retroauricular approach was performed. From the discussion (Line 451 -452) I was confused as here it just speaks about not inserting a CI.
    • Where were the recordings made in the left or right AC, or both? In other words: ipsi or contralateral to the CI or both?
    • There is no explanation of the CF shift assessment as being a shift or not.
  2. Based on the presented data I wonder whether there were CF shifts that were statistically missed. CI implant animals show shift towards lower CFs at higher CF regions (> 4.5 kHZ, -0.4 to -0.8 Oct). In contrast control animals do show comparably large shifts at lower freqs towards higher CFs (<4.5 kHz, 0.3 Oct). If true, then is it conceivable that the CI implantation actually shifts CFs upwards in lower CF regions by 0.45 Oct?
  3. I am not convinced yet that using suprathreshold levels to eliminate effects of threshold on firing rate is sufficient. Usually rate-level functions for unit activity are monotonic in AC under anesthesia. Thus a simple choice of suprathreshold level (75 dB) might not suffice to eliminate the effect of firing rate changes due to increase in threshold. Also there is no analysis of potential changes in spontaneous firing rate. I find these results particularly interesting as they might speak towards rapid changes in rate-level functions with potentially increasing slopes that could differ between frequency regions. In turn, this could provide clues about what is changing due to CI insertion.

Therefore, I suggest to provide these analyses.

  1. Discussion: I suggest to discuss the expected frequency regions where the CI is located within the cochlea of the guinea pig. This would allow for a more comprehensive understanding of the effect the mechanical damage in general to the bulla and cochleostomy versus changes due to inflammation and restriction of basilar membrane movement due to CI insertion.

If the effect of CI insertion remains restricted to the frequency region where the CI is located then clinically this might not be relevant as the surgeon will aim to fit into the deafened region. Also, the changes have been evaluated in acute experiments. Thus the data do not by themselves speak to whether this has clinical relevance as the threshold shifts might be temporary.

Consequently, I suggest to improve the section on clinical implications.

Minor points:

Line 111 – 112: The supplemental doses are not explained sufficiently. How does 0.3 – 0.5 ml correspond to a urethane dosage?

Line 223 – 224: The CF does not always shift towards lower frequencies. In fact, if I just compare animal 1 and 2 the shifts seem comparable but for CH3 and 4 in animal 2. The description here thus does not seem to fit the presented data and caption of figure 1.

What criterion was used to assess something to be ‘a shift’ or not?

Line 231 - 232: Delete ‘that’ from: ‘From each electrode that there…’ and ‘characteristic frequency' as CF was introduced as an abbreviation already.

Line 238: Labels for figure 1 could be improved. They are very hard to read. For animal A there are no abscissa labels but in B and C there are.

Line 248: 9/16 does not seem to be ‘many’ if the rest shift towards higher values.

Line 252: same as above: how was it determined that a CF did or did not shift?

Line 266 ff: are the quantitative data given as mean +- SEM?

Line 267: the CFs in figure 1 do not seem to shift by 0.8 octaves on average for the high CFs. Maybe the authors can identify an example that better suits the average data?

Line 256 ff: are the shifts in the chosen frequency regions themselves significant? I am thinking about the analysis strategy employed for threshold shifts vs. frequency but using CF distances as the observable.

Line 274 – 275: how where the additional insertions, mentioned in the methods, integrated and tested here?

Line 288: Labels are hard (A, B) or even impossible to read (C, D). Please improve. I suggest to also add lines to indicate the frequency ranges chosen for statistical analysis in A, B.

Line 349: evoked firing rate should be introduced in the methods section. Which time windows were used for spontaneous rate and stimulus related firing evaluation?

Line 377 – 380: I suggest to make it clear that currently the summary sentence refers to the evoked firing rate.

Author Response

see attached document

Round 2

Reviewer 2 Report

I would like to thank the authors for a nice scholarly discussion. The authors have sufficiently addressed my concerns. Thus, I have only very small things left.

Minor points:

There must have been some problem with the figures in the new pdf manuscript version. The authors improved labels in figure 1 and 2 as well as added another figure 5. However, figure 1 and 2 look the same as the original version if I plot them on top of each other and change transparency. Also, there are only 5 figures in my version for review. Probably just a conversion issue.

Consequently, I could not evaluate the rate-level functions presented in the added figure 5.

In the supplementary word document the figure 5 caption: “B. In this case, the threshold (defined as a response above spontaneous activity plus 6 sem) increased from XX to VV dB” is not complete yet.

Specify in the text that the data are mean +- SEM. E.g. line 270 ff.
